

# Ultrasonic Nebulization for the Elemental Analysis of Microgram-Level Samples with Offline Aerosol Mass Spectrometry

Rachel E. O'Brien[1*], Kelsey J. Ridley[2], Manjula R. Canagaratna[3], John T. Jayne[3], Philip L. Croteau[3], Douglas R. Worsnop[3], Sri Hapsari Budisulistiorini[4†], Jason D. Surratt[4], Christopher L. Follett[5], Daniel J. Repeta[5], Jesse H. Kroll[2]

[1] Department of Chemistry, College of William and Mary, Williamsburg, Virginia, 23185, USA
[2] Department of Civil and Environmental Engineering, Massachusetts Institute of Technology, Cambridge, Massachusetts 02139, USA
[3] Center for Aerosol and Cloud Chemistry, Aerodyne Research Inc., Billerica, Massachusetts 01821, USA
[4] Department of Environmental Sciences and Engineering, Gillings School of Global Public Health, University of North Carolina at Chapel Hill, Chapel Hill, North Carolina 27599, USA
[5] Department of Marine Chemistry and Geochemistry, Woods Hole Oceanographic Institution, Woods Hole, Massachusetts 02540, USA

† Now at: Earth Observatory of Singapore, Nanyang Technological University, Singapore 638789, Singapore

Corresponding author: Rachel O'Brien, reobrien@wm.edu

**Abstract.** The elemental composition of organic material in environmental samples – including atmospheric organic aerosol, dissolved organic matter, and other complex mixtures – provides insights into their sources and environmental processing. However, standard analytical techniques for measuring elemental ratios typically require large sample sizes (milligrams of material or more). Here we characterize a method for measuring elemental ratios in environmental samples, requiring only micrograms of material, using a Small Volume Nebulizer (SVN). The technique uses ultrasonic nebulization of samples to generate aerosol particles (100-300 nm diameter), which are then analyzed using an aerosol mass spectrometer (AMS). We demonstrate that the technique generates aerosol from complex organic mixtures with minimal changes to the elemental composition of the organic and that quantification is possible using internal standards (e.g., $NH_4^{15}NO_3$). Sample volumes of 2-4 µL with total solution concentrations of at least 0.2 g/L form sufficient particle mass for elemental ratio measurement by the AMS, despite only a small fraction (~0.1%) of the sample forming fine particles while the remainder end up as larger droplets. The method was applied to aerosol filter extracts from the field and laboratory, as well as to dissolved organic matter (DOM) from the North Pacific Ocean. In the case of aerosol particles, the mass spectra and elemental ratios from the SVN-AMS agree with those from online AMS sampling; similarly, for DOM, the elemental ratios determined from the SVN-AMS agree with those determined using combustion analysis. The SVN-AMS provides a platform for the rapid quantitative analysis





of the elemental composition of complex organic mixtures and non-refractory inorganic salts from microgram samples
with applications that include analysis of aerosol extracts, and terrestrial and atmospheric dissolved organic matter.
**1 Introduction**

A large number of environmental systems, including the atmosphere, natural waters, and terrestrial systems,

contain complex organic mixtures composed of hundreds to thousands of molecular species.  Our ability to understand
and model such complex chemical systems is often greatly improved when we characterize them in terms of simple
chemical frameworks. On the simplest level, the analysis of average elemental ratios can provide important
information on potential sources of organic matter samples, as well as the chemical and/or biological transformation
processes that modify their composition. For example, the elemental ratios of atmospheric organic aerosol – e.g.,
oxygen/carbon ratio (O:C), hydrogen/carbon ratio (H:C), and nitrogen/carbon ratio (N:C) – provide information on
aerosol sources and aging (Aiken et al., 2008; Canagaratna et al., 2015; Chen et al., 2015; Daumit et al., 2013; Heald
et al., 2010; Jimenez et al., 2009; Kroll et al., 2011). Similarly, in water and soil samples, the elemental ratios of
carbon, nitrogen, and phosphorous reveal insights into sources and processing of dissolved and particulate organic
matter (Becker et al., 2014; Hansman et al., 2015; Koch et al., 2005; Lu et al., 2015).

The most widespread technique for elemental analysis is high-temperature combustion followed by elemental

(CHNS) analysis, which is highly accurate but can require milligrams of material (Skoog et al., 1998).  For many trace
environmental samples, like atmospheric aerosol, this can require extremely long collection times which lead to low
time resolution, limiting the amount of information provided for systems that exhibit high temporal variability.  An
alternative approach for measuring the elemental ratios of aerosol is online (real-time) techniques. The most widely-
used instrument for such measurements is the Aerodyne High-Resolution Time-of-Flight Aerosol Mass Spectrometer
(HR-ToF-AMS) (Decarlo et al., 2006), which can measure elemental ratios of ambient aerosol using just nanograms
of material.  Over the last decade, in-situ analysis of aerosol particles with the AMS has enabled rapid, sensitive
characterization of aerosol concentrations, sources, and atmospheric aging, improving our ability to model
atmospheric aerosol and consequently its climate and health effects (Kroll et al. 2015;  Ng et al. 2011; Jimenez et al.
2009; Canagaratna et al. 2007).

Recently, a number of researchers have used the AMS in an "offline mode," in which atmospheric samples

are collected on filters, extracted, and then atomized into the AMS.  Examples include the analysis of sources and
aging of atmospheric organic material from aerosol filter extracts (Bozzetti et al., 2017; Huang et al., 2014; Sun et al.,
2011; Xu et al., 2015; Ye et al., 2017), cloud/fog water samples (Kaul et al., 2014; Lee et al., 2012), and organic
material in glaciers (Xu et al., 2013). Offline AMS has proven especially useful for the analysis of aerosol particles
larger than 1 μm (Bozzetti et al., 2016; Daellenbach et al., 2016; Ge et al., 2017).  Offline AMS has also proven useful
in investigating fractionation and solubility of atmospheric organic material in water and organic solvents
(Daellenbach et al., 2016; Mihara and Mochida, 2011). These studies used both custom-made and commercial
atomizers with solvent volumes of at least 5-15 mL. To generate aerosol particles in the size range needed for the
AMS, this corresponds to necessary sample masses on the order of 50 μg.  While this represents a substantial
improvement over the sample mass requirements of conventional CHNS analysis, it is still sufficiently large to limit



the applicability of the approach since it can require relatively large organic samples collected with high-volume
samplers, often over 24 hours or more.

In this work, we characterize a new technique for the elemental analysis of very small sample masses, using

ultrasonic nebulization. Aerosol generation with a small volume nebulizer (SVN) expands the range of environmental
samples that can be measured, where either sample size is limited or solvent contamination is a concern.  The SVN
generates aerosol suitable for analysis with aerosol instrumentation, including not only the AMS but also Scanning
Mobility Particle Sizers (SMPS); single particle mass spectrometers (e.g. Particle Analysis by Laser Mass
Spectrometry (PALMS) (Murphy et al., 1998)); soft ionization sources (e.g. Extractive Electrospray Ionization (EESI)
(Gallimore and Kalberer, 2013)); and thermal desorption chemical ionization mass spectrometers (e.g. Filter Inlet for
Gases and AEROsols, (FIGERO-CIMS) (Lopez-Hilfiker et al., 2014)).  Here, we present results characterizing the
SVN using an HR-ToF-AMS and an SMPS and demonstrate production and elemental analysis of aerosol using 2-4
µL of liquid samples, with masses of organic material as low as ~0.4 µg. Quantification of total organic concentrations
is demonstrated using internal standards.  We examine the effects of aerosol collection, extraction, and nebulization
on the mass spectra and elemental ratios observed for offline and online AMS.  The aim of this work is to demonstrate
that offline analysis of organic mixtures with the SVN-AMS can provide quantitative characteristic elemental ratios
for trace environmental and biological samples using just micrograms of sample.
**2 Experimental**
**2.1 Small Volume Nebulizer**

The SVN, shown in Figure 1, creates an aerosol by ultrasonically nebulizing a small droplet placed on a thin

film stretched across a water reservoir.  The aerosol is then carried by a gentle flow of either house air or argon (Airgas,
99.999% purity) into the AMS. The three main components of the SVN, described in detail below, are (1) a bottom
cylinder with an ultrasonic transducer and water bath, (2) a thin film that is press-fit onto the top of the water bath by
an upper cylinder with a slightly larger ID, and (3) a vertical glass tube that connects to the AMS.  The connections
between all components are airtight, but the apparatus is easily disassembled to inject samples onto the film, as well
as to clean the thin film and change the water bath.

In the bottom section of the SVN, the 2.4 MHz ultrasonic transducer (Sonaer, Inc., Model 241VM) is located

just under the liquid reservoir, with a thin film stretched across the top of the reservoir to provide a clean nebulization
surface for the sample.  We use a 0.001" thick Kapton film or Teflon film, as these two were found to have the lowest
background signal and the best performance in terms of the amount of aerosol generated compared to other materials
tested. Press-fit onto the bottom piece is another PVC cylinder that has two side ports with carrier gas inlets, and a
larger hole in the top into which a 15 cm glass tube is seated. The distance from the thin film to the bottom of the glass
tube is ~1.5 cm. During experiments, the nebulized aerosol is carried up through the vertical glass tube, into the
stainless steel tubing that leads to the AMS.  Additional components such as Nafion™ (Perma Pure LLC) driers can
be placed inline if desired, but such modifications were not investigated in the present work.



107 Samples can be introduced into the SVN using two different approaches: discrete injections of individual

108 samples (for individual "one-shot" measurements) or continuous addition of a sample flow (for continual analysis,

109 enabling signal averaging). For most studies, MilliQ water was used as the solvent; in some cases we used HPLC-

110 grade methanol, though the organic background signal is higher in that case, likely due to a combination of increased

111 organic background in organic solvents and incomplete evaporation of methanol prior to measurement. For most of

112 the work described here, we used discrete injections of 2-5 µL of aqueous solutions manually deposited directly onto

113 the center of the Kapton film. For continuous injections, solutions made with MilliQ or organic solvents were

114 introduced via a syringe pump (Harvard Apparatus Model 22), which sends liquid flow (20-40 µL/min) through a

115 borosilicate capillary entering the SVN via a small downward-facing hole in the upper PVC piece (Figure 1). In the

116 future, such a port could also be used to provide automated discrete sample introduction using an autosampler.

117 For aqueous samples containing salts and small organic molecules, only 1-2% of the original sample mass

118 was observed to remain on the thin film after a discrete injection (Figure S2). To ensure a clean surface between

119 different samples, the surface was cleaned by nebulizing 2-8 µL of MilliQ water off the surface 5-10 times over

120 approximately one minute. The cleanliness of the surface was then verified by running a salt solution (at least 0.5

121 g/L) between each sample. The salt solution is necessary to ensure that any contaminants can be seen, since pure water

122 risks generating aerosol particles that are too small to be measured in the AMS. For samples in which carryover was

123 observed (for example, the dissolved organic matter solutions discussed in section 3.1), additional cleaning of the film

124 was undertaken with sonication in a deionized water bath followed by rinsing with HPLC-grade methanol for > 30

125 seconds. Careful maintenance of the surface ensures uncontaminated mass spectra and accurate quantification of the

126 solution components.

127 **2.2 AMS Data Collection and Analysis**

128 While a number of different aerosols instruments could be used with the SVN, here we focus primarily on

129 elemental analysis by the HR-ToF-AMS. The AMS has previously been described in detail (Canagaratna et al., 2007;

130 Decarlo et al., 2006) and provides quantitative measurements of non-refractory material (organics, ammonium sulfate,

131 ammonium nitrate, etc.) for aerosol particles between approximately 40 and 1,000 nm. The mass spectrometer used

132 in the AMS is a high resolution time-of-flight mass spectrometer (HTOF-MS, Tofwerk AG), run under "V mode" for

133 a mass resolution of 2,000-3,000 m/Δm. This mass-resolving power enables peak fitting and identification of all

134 organic fragment ions observed here (< 130 $m/z$), which enables the calculation of quantitative elemental ratios for

135 the organic mixture, after correcting for fragmentation bias during electron ionization (Aiken et al., 2007, 2008;

136 Canagaratna et al., 2015). For AMS data collected using indoor or outdoor air, the intensities of $CO^+$ and $H_2O^+$ are

137 complicated by gas-phase interferences ($N_2^+$ and gas-phase $H_2O^+$). For samples compared to chamber or ambient

138 online-AMS data sets, house air was the carrier gas and standard empirical estimates were used (Canagaratna et al.,

139 2015). With the SVN, inert carrier gases such as argon can also be used, allowing for the direct measurement of the

140 $CO^+$ ion intensity (as demonstrated below for dissolved organic matter).

141 For discrete sampling, "fast MS" mode (Kimmel et al., 2010) was used because the pulse length of a single

142 injection is ~30-60 seconds long. Fast MS mode generates mass spectra every 0.5-2 seconds and the instrument cycles



between the "closed" state, in which the aerosol beam is blocked, and the "open" state, in which the aerosol beam can
reach the vaporization/ionization region for detection.  For the work shown here, mass spectra were collected every
0.5 seconds for ~15-18 seconds in the "open" state, followed by 3 seconds in the closed state.  The closed spectrum
provides information on the instrument background, including contributions from gas phase species, and is subtracted
from the open spectrum in data processing.  For continuous injections, the standard AMS operating mode ("GenAlt
mode") was used. This provides an average mass spectrum (by subtracting the closed signal from the open signal), as
well as particle time-of-flight (PToF) data (providing aerosol size distributions for all aerosol components), once per
minute. All AMS data were analysed using software packages SQUIRREL (v1.57I) and PIKA (v1.16I), available at
http://cires1.colorado.edu/jimenez-group/ToFAMSResources/ToFSoftware/.

The aerodynamic lens on the AMS has a transmission efficiency of nearly 100% for particles with

aerodynamic diameters of 70-500 nm; for somewhat smaller particles (30-70 nm), this transmission is lower but not
negligible  (Jimenez et al., 2003).  Thus, high enough solution concentrations are used such that the dried particles
formed in the nebulizer are larger than ~100 nm aerodynamic diameter.  Collection efficiencies (CE) in the AMS can
vary depending on the extent to which aerosol particles bounce off the thermal element prior to vaporization.  This
can impact the absolute concentrations observed, but for internally mixed samples, the relative concentrations of
different aerosol components are independent of CE.  In this work, most measurements (including elemental ratios)
are reported as relative measurements, and thus no CE correction is applied.  Some biases may arise if the aerosol is
not internally mixed, but for all systems examined so far in PToF, no size-dependence in composition was observed
(Figure S1).

**2.3 Sample Collection and Solution Preparation**

As described below, samples were prepared from a number of sources, including commercially available

standards, the extracts of chamber and ambient aerosol particles collected on filters, and dissolved organic matter from
the Pacific Ocean. For all solutions, either ultrapure water (18.2 M $\Omega$ cm, MilliQ) or HPLC-grade methanol was used.
Prior to use, all glassware was cleaned with a methanol solvent wash and baked at 450°C for 6 hours.

Chamber aerosol (enabling offline vs. online comparisons) was generated in the MIT 7.5-m$^3$ Teflon

environmental chamber.  Details on the facility are given elsewhere (Hunter et al., 2014). Experiments were run at 20
°C, < 5% RH, in the dark, and under low-NO$_x$ conditions using ozone as the oxidant.  Ammonium sulfate seeds were
added to a concentration of ~60 $\mu$g/m$^3$.  The VOC, $\alpha$-pinene, had an initial mixing ratio of 100 ppb; a penray lamp
(Jelight model 600) was used to add ~500 ppb ozone. Filter samples were collected on Zeflour® PTFE Membrane
Filters (0.5 $\mu$m pore size) at flow rates of ~5 L/min for 10 hr.  Laboratory blank filters were prepared by placing
separate filters in the filter holder for 10 minutes before the start of the experiments.  All filters were stored in baked
aluminum foil packets, sealed in plastic bags, and placed in a freezer at -20 °C until extraction. Filters were extracted
with ~4 mL of HPLC-grade methanol.  In order to avoid oxidation of the organic species in the extract, no sonication
was used; instead, the vials were gently agitated by hand intermittently over 3 hours.  Solutions were concentrated by
evaporating to dryness under a gentle stream of ultra-high purity N$_2$.  Dried samples were stored in the freezer at -20





°C until reconstitution with MilliQ water and analysis by the SVN-AMS. Blank subtraction was carried out with a
scaling of the filter blank to 12% of the sample signal, as determined from the internal standard in each sample.
Field samples from the Southern Oxidant and Aerosol Study (SOAS) in 2013 were collected on pre-baked
Tissuquartz™ Filters (Pall Life Science, 8 x 10 in) at Look Rock, TN starting on 06/16/2013 using a high-volume
aerosol filter sampler with a $PM_{2.5}$ cyclone (Tisch Environmental, Inc.) as described by Budisulistiorini et al. (2015).
For filter extraction, a 37 mm punch was extracted in a pre-cleaned scintillation vials with 20 mL high-purity methanol
(LC-MS Chromasolv-grade®, Sigma Aldrich) by sonication for 45 min. Filter extract was filtered through 0.2 µm
syringe filter (Acrodisc® PTFE membrane, Pall Life Sciences) to remove suspended filter fibers. The filtered extract
was then blown down to dryness under a gentle $N_{2(g)}$ stream at room temperature. An aerosol chemical speciation
monitor (ACSM) (Ng et al., 2011a) was deployed at the same field site (Budisulistiorini et al., 2015); the average mass
spectrum for the length of the filter sample was used for comparison with the present SVN-AMS measurements.
Standard solutions were prepared from commercially available compounds dissolved in MilliQ water.
Reagents used included ammonium sulfate, ammonium nitrate, isotopically-labelled ammonium nitrate ($NH_4^{15}NO_3$),
citric acid, mannitol, PEG-400, 4-hydroxy-3-methoxy-DL-mandelic acid (HMMA), and HPLC grade methanol, all
from Sigma-Aldrich.
The DOM was collected at the Natural Energy Laboratory Hawaii Authority facility in Kona, Hawaii.
Seawater from a depth of 20 m was pumped though a 0.2 µm filter to remove particles and the high molecular weight
fraction of organic matter in the filtrate was concentrated by ultrafiltration using a membrane with a 1 nm pore size
and a nominal 1,000 Dalton molecular weight cut off. This fraction was desalted by serial dilution/concentration with
MilliQ water and then freeze-dried. Low-molecular weight humic substances and residual salts were removed by
stirring with anion (hydroxide form) and cation exchange resins (hydrogen form). The final product was freeze-dried
to yield a fluffy white powder. Conventional CHNS analysis was carried out using a CE-440 Elemental Analyzer
(Exeter Analytical).
**3 Results and Discussion**
**3.1 Nebulization and Aerosol Size**
Figure 2a shows a time series of measured aerosol mass concentrations of a typical nebulized aerosol pulse
from a 4 µL solution containing approximately 0.33 g/L each of mannitol, ammonium sulfate, and ammonium nitrate.
The nebulizer is turned on at t = 0 and shortly afterwards (t = ~10 s) the aerosol packet is observed in the AMS. The
start of the nebulization is timed so that a closed (background) measurement occurs during the downslope of the signal
(t=~16-21 s, dashed lines). This background is subtracted from the aerosol particle signal during data processing.
Measurements are collected until the signal returns to the baseline (t=~44 s).
Figure 2b shows the size distribution of the particles generated by nebulizing an aqueous solution of citric
acid with continuous injection via syringe pump and a total concentration of ~1 g/L into an SMPS (TSI). The particles
have size distributions centered at 150-200 nm. We find injections of solutions with total concentrations above 0.2
g/L provide sufficient aerosol mass for analysis (Figure S1). These measurements compare well with calculations



based on the size of droplets reported by the manufacturer (Sonaer inc.) of approximately 1.7 μm using water solutions.
Assuming that the density of the dried particle is 1.3 g/cm$^3$ (Nakao et al., 2013), the minimum sample concentration
that will form a 100 nm dried particle is approximately 0.3 g/L. To generate large enough aerosol particles from more
dilute solutions, larger initial droplets could be formed by changing the piezo to a transducer that vibrates at a lower
frequency. However, for these larger droplets, drying will require the loss of a greater amount of solvent, so that any
impurities in the solvent will make up a larger (and possibly even dominant) fraction of the resulting fine particles.
Thus the use of ultrasonic nebulization at lower frequencies was not investigated here.

## 3.2 Quantification

### 3.2.1 Nebulization Efficiency

A key quantity describing the potential sensitivity of the SVN-AMS is the SVN nebulization efficiency, the
ratio of the mass measured in the AMS compared to the mass of analyte placed on the thin film. This was determined
by loading 4 μL of a known solution onto the film and measuring the mass of each component in the AMS integrated
over the injection pulse, determined by:
$$M_{AMS} = \int_{t_1}^{t_2} f(t)dt \times v_{AMS}$$

where $M_{AMS}$ is the mass measured by the AMS in μg, $f(t)$ is the instantaneous mass concentration measured in the
AMS (μg/m$^3$), and $v_{ams}$, is the gas flow rate into the AMS in m$^3$/s. For each injection, the background-subtracted AMS
signal is calculated (Figure 2a). The gaps due to closed cycles are bridged by interpolation and the area under the
injection curve is calculated via trapezoidal integration from time points before and after the pulse ($t_1$ and $t_2$,
respectively) with the time steps (dt) corresponding to the MS cycle time (here 0.5 s). The mass measured in the AMS
is affected by three factors: the amount of aerosol formed and transported out of the SVN, the fraction of the gas flow
from the SVN that is sampled by the AMS (typically ~1/2), and the fraction of aerosol that bounces off the heater
element before vaporizing (the AMS CE).
Figure 3 shows the mass measured in the AMS compared to the mass deposited on the nebulizer for replicate
injections of four different aqueous solutions of citric acid, ammonium nitrate, ammonium sulfate, and isotopically-
labeled ammonium nitrate (NH$_4$$^{15}$NO$_3$, used later as an internal standard) with concentrations ranging between
approximately 0.1 and 0.2 g/L for each of the components (but with the same total concentration, 0.75 g/L). The
amount of mass measured in the AMS increases slowly compared to the amount placed on the film, and variations in
measured masses are observed for replicate injections of the same sample. The observed increase in the mass
measured for these samples is likely partially related to CE on the vaporizer, as the highest efficiency was observed
for samples with the largest mass fractions of organic. The measured nebulization efficiencies are on the order of
0.02-0.06%, indicating that the aerosol mass detected with the AMS is approximately three orders of magnitude lower
than the mass originally deposited on the thin film.
The majority of the sample mass loss likely occurs during the nebulization process itself. For aqueous
solutions in the SVN, large droplets are observed to be ejected off the surface of the film at the same time as the
aerosol is generated. These ejected droplets are then lost to the walls of the SVN. The ejection of these droplets





appears to be a necessary part of the nebulization mechanism for water samples as smaller volumes (< 1 µL) of water
do not generate such droplets and also do not appear to form aerosol. This observed mechanism is in agreement with
previous studies of aerosol generation for ultrasonic nebulization, in which cavitation within the droplet (Lang, 1962)
and boiling and/or jetting from a droplet chain (Simon et al., 2015) have been observed.
The size distribution and number of aerosol particles from ultrasonic nebulization have been shown to be
affected by the frequency of the ultrasonic vibration, the properties of the liquid including surface tension, density,
and viscosity, and the concentration of the solution (Donnelly et al., 2005; Lang, 1962; Simon et al., 2015). The present
application involves relatively dilute solution, so the only parameter that could be varied was the surface tension, by
use of different solvents. Nebulization of solvents with lower surface tension, such as methanol, led to the ejection of
much smaller droplets, and consequently substantially higher nebulization efficiencies (~10%). However, methanol
(and other HPLC-grade organic solvents) was found to give higher background signals in the AMS than MilliQ water,
likely due to higher levels of low-volatility contaminants. This difference was also observed by Daellenbach *et al*.
(2016); therefore, MilliQ water appears to be the ideal solvent to use for most environmental samples. However, with
adequate solvent background characterization, organic solvents may be optimal for environmental samples with more
non-polar components (e.g. petroleum or fresh tail pipe emissions).
**3.2.2 Internal Standards and Calibration Curves**
In Figure 3, the vertical spread of data points shows the variation in nebulization efficiency from one injection
to the next. This is likely the result of small differences in the droplet shape or position on the film, leading to
differences in how the droplets are ejected from the surface during aerosol formation. This run-to-run variability in
nebulization efficiency, as well as the lack of a linear correlation between the mass placed on the film and the mass
observed, complicates quantification, and necessitates the use of an internal standard to quantify the concentration of
organic species within the original sample. In some cases, an inorganic ion that is independently quantified, such as
sulfate, can serve as this internal standard (Daellenbach et al., 2016). However, in many cases such an independent
measurement is not available; additionally, some environmental samples may not contain appreciable levels of
measurable inorganic species, or else such species may not be soluble in the solvent of choice (e.g. ionic species in
organic solvents). In these cases, an internal standard needs to be added to the solution prior to nebulization.
For use with the AMS, the internal standard must meet a number of requirements: it must be non-refractory,
soluble, unreactive with the other sample components, not already present in the solution, and easily distinguishable
from other species in the sample. For nebulization of samples dissolved in organic solvents, organic internal standards
(e.g., phthalic acid (Chen et al., 2016; Han et al., 2016)) meet these requirements. In the present work, which focuses
on aqueous samples only, we use an inorganic internal standard of isotopically-labelled ammonium nitrate
($NH_4{}^{15}NO_3$). An example mass spectrum for an internal standard solution is shown in Figure 4a. The background
signal from other components (organic, sulfate, and nitrate) is very low. Another tested option is ammonium iodide
($NH_4I$). Both of these salts work well as internal standards for both laboratory and ambient samples, since neither
$^{15}NO_3$ nor iodide are present in appreciable amounts in the atmosphere and there is usually a very small contribution


of organic fragments at the fragment masses observed for those salts.  The internal standards are added at the same
order of magnitude concentration as the sample.

Figure 4b shows calibration curves with linear responses for three different organic compounds (citric acid,

4-hydroxy-3-methoxy-DL-mandelic acid (HMMA), and polyethylene glycol 400 (PEG-400)) at four concentrations
using $NH_4{}^{15}NO_3$ as the internal standard.  For the calibration curve, the ratios of the AMS signals for the analyte over
the internal standard are compared to the ratios for known solution concentrations, thus correcting any variations in
the mass of analyte nebulized.  For quantification of unknowns, known concentrations of the internal standard are
added to the samples. The ratio of the measured AMS signals can then be used to calculate the unknown analyte
concentration from the calibration curve.

For quantification of complex organic mixtures using this technique, the most accurate organic calibration

standards will have chemical structures similar to the average structure of the mixture.  The slope of each line is related
to the relative ionization efficiency (RIE) of the organic compound in the AMS (Jimenez et al., 2003). The RIE values
in Figure 4b for HMMA and citric acid (1.01 and 1.95, respectively) bracket the range of RIE values for different
types of organics measured using standard AMS calibration techniques (Jimenez et al., 2016).  This range likely arises
from differences in how the organic compounds dissociate during volatilization on the heater. The heater in the AMS
is typically set at 600°C, and so most organic molecules found in organic aerosol thermally decompose prior to electron
impact ionization (Canagaratna et al., 2015) leading to RIEs in the range of 1.0-2.0.  In contrast, the slope of 2.62 for
PEG-400 is substantially outside of the range of values. However, with the AMS, complex mixtures are less likely to
show large variations in RIE than different individual compounds, such as those used in Figure 4.  For extracts of
atmospheric aerosol or other smaller organic mixtures, the RIE of 1.4, which is typically used for AMS measurements
(Canagaratna et al., 2007; Jimenez et al., 2016; Xu et al., 2018), is likely the best value to use.  For extracts of other
types of organic mixtures, compounds that have a structure similar to the average organic composition should be used
to calibrate the samples.

## 3.1 Mass Spectral Analysis

The primary goal of the SVN-AMS is to measure quantitative chemical information, specifically elemental

ratios, from complex organic mixtures. We have characterized these for a number of different chemical systems,
described below. Results are summarized in Figure 5 (comparing SVN-AMS and online AMS mass spectra) and Table
1 (comparing elemental ratios measured with SVN-AMS with those measured by either online AMS or CHNS
analysis).

One concern with using ultrasonic nebulization to generate aerosol particles is the possibility that the high

temperatures possibly reached by the solution during nebulization may degrade the organic compounds, affecting their
mass spectra (and hence measured elemental composition). Figure 5a shows a comparison of a solution containing 1
g/L citric acid aerosolized with a TSI atomizer (black) and the SVN (green), with the inset showing a direct comparison
between the intensities measured for each ion in the mass spectrum.  The degree of agreement can be described by the
dot product of the two spectra, as well as the log of the intensities before taking the dot product (log-dot product),
which gives the lower intensity peaks greater weight.  Very good overlap between the two mass spectra is observed,





with a dot product of 0.99 and a log-dot product of 0.96. This indicates minimal degradation of the citric acid by
ultrasonic nebulization.

A high degree of similarity is also observed between offline and online aerosol measurements for more
complex mixtures. Figure 5b shows mass spectra for a comparison of offline (red) vs. online (black) SOA sample,
generated from the dark ozonolysis of α-pinene. For all filter samples, spectra from the SVN are background
subtracted using spectra collected from blank filter samples. The overlap in Figure 5b between the mass spectra is
very good, with a dot product of 0.98 and a log-dot product of 0.98. The elemental ratios are also very similar between
the two samples with an H:C of 1.6 for both and O:C of 0.48 for the chamber and 0.49 for the SVN samples (Table
1). The largest difference is observed at $m/z$ 44 ($CO_2^+$) and $m/z$ 43 ($C_2H_3O^+$) with a larger fraction of $CO_2^+$ in the
offline sample. The intensity of $CO^+$ ($m/z$ 28) is also different, but only because it is set equal to the intensity of the
$CO_2^+$ ion, as is commonly done for ambient sampling with the AMS (given that the $CO^+$ ion generally cannot be
distinguished from the much more abundant $N_2^+$ ion (Aiken et al., 2007). The organic contribution from $H_2O^+$, $OH^+$,
and $O^+$ is also constrained by the $CO_2^+$ signal so any differences in $CO_2^+$ intensity will also show up in those ions
(Aiken et al., 2008). The observed difference in $CO_2^+$ and $C_2H_3O^+$ ion intensity is likely a result of the extraction step
prior to nebulization, which may preferentially dissolve the most water-soluble (oxidized) SOA components; however,
based on the agreement in H:C and O:C in the online and offline cases, this does not appear to bias elemental ratio
measurements substantially.

Figure 5c shows a comparison of online and offline measurements of ambient organic aerosol, specifically
ACSM measurements and SVN-AMS measurements of a filter extract collected simultaneously during the 2013
SOAS field campaign in Look Rock, TN (8 pm July 4 to 7am July 5, 2013; EST). Since the ACSM is a unit-mass-
resolution instrument, the HR-AMS data are degraded to unit mass resolution, and ions that are determined from the
$m/z$ 44 signal ($m/z$=15, 16, 17, 18, and 28) are excluded from the analysis. Additionally, ions at $m/z$ 30 and 31 were
removed from comparison because of interferences from the internal standard ($m/z$ 31) and nitrate in the sample ($m/z$
30).

The two mass spectra in Figure 5c have a high degree of agreement between the major ions (dot product of
0.98). However, there is substantially more variation between the two techniques than in the chamber study, especially
in the lower-abundance peaks ($m/z$>45; see inset), as reflected in the lower log-dot product of only 0.90. Possible
reasons for this lower correlation include fractionation from the extraction step, the different sizes measured ($PM_{2.5}$
for the filter vs. $PM_1$ for the ACSM) (Daellenbach et al., 2016), the uncertainty in ACSM signals at higher masses due
to uncertainty in the relative ion transmission curve (Ng et al., 2011a), and/or the losses of more volatile compounds
during collection, extraction, and handling. Additional work is necessary to quantify the importance of these effects.
Regardless, the high degree of overlap between the online (AMS/ACSM) measurements and offline (SVN-AMS)
results indicates that the ensemble organic composition for these aerosol samples is generally well-represented by the
SVN-AMS measurements (Table 1).

For the SVN, the small sample volume requirements can make it attractive for the analysis of other
environmental samples that are soluble in water (or organic solvents) and that have low enough vapor pressures to
remain in the condensed phase after nebulization. Figure 3d shows an example AMS mass spectrum from dissolved



organic matter (DOM) from the Pacific Ocean. The mass spectrum is dominated by oxidized fragments containing
one or more oxygen atoms with smaller amounts of nitrogen-containing fragments. The measured N:C and H:C values
of 0.081 and 1.7, respectively, matches those measured by CHNS analysis (0.080 and 1.74, respectively). This
demonstrates that with the SVN, microgram quantities of dissolved environmental mixtures can be nebulized and
sampled into the AMS providing a rapid, quantitative method to determine elemental ratios in these complex organic
mixtures.
**4 Conclusions**
A new ultrasonic nebulizer has been described and characterized for generation of aerosol from very small
sample masses. We demonstrate the application of this technique to offline AMS analysis of complex organic
mixtures from aerosol filter extracts and DOM. Data sets that include quantitative organic mass, characteristic mass
spectra, and quantitative elemental ratios can be generated from only 0.4-1.2 µg of material. A direct comparison
between the mass spectra generated by commercial spray atomizers or by real-time aerosol particles sampled directly
from the atmosphere showed high degrees of agreement. Nebulization of aqueous samples generated measureable
aerosol from 0.1% of the sample mass. Higher nebulization efficiencies (and smaller ejected droplets) were observed
for methanol, likely due to its lower surface tension. The SVN, combined with offline-AMS, provides rapid analysis
of non-refractory organic and inorganic compounds. For other types of characterization, including analysis of
refractory material or organic molecular composition, the SVN can also be coupled with other aerosol instrumentation
such as PALMS or CIMS instruments.
Future improvements in the nebulization and collection efficiency of the SVN-AMS will enable analysis with
even lower sample mass requirements. The use of organic internal standards is one method to potentially improve
collection efficiency in the AMS. Additionally, the use of solvents with lower surface tension than water shows
promise for improved nebulization efficiencies. A useful future direction for this technique will be to characterize the
background signal in different organic solvents and optimize the continuous flow configuration to minimize the return
of large ejected droplets back onto the film. Continuous flow with organic solutions will also enable the analysis of
more hydrophobic organic samples such as fresh vehicle emissions, cooking oils, and petrochemical samples. In the
future, the SVN can be used to generate aerosol for analysis of other environmental samples to investigate sources or
processing/aging of these organic mixtures. The SVN, combined with aerosol measurement techniques such as the
AMS, provides a rapid, quantitative method to characterize the chemical and elemental properties of complex organic
mixtures, producing rich data sets for the exploration of exceptionally trace environmental samples.



**Supporting Information**

The supporting information is available free of charge at DOI:xxx. The document contains additional information on particle sizes and memory effects between runs, (file type, PDF).

**Corresponding Author**

Rachel E. O'Brien, College of William and Mary, reobrien@wm.edu.

**Author Contributions**

MRC, JTJ, PLC, DRW, JHK, and KJR designed and built the SVN. SHB, JDS, CLF, DJR provided ambient aerosol samples and DOM. REO and JHK designed experiments and REO carried them out. REO prepared the manuscript with contributions from all authors.

**Data availability**

All data sets including mass spectra and SMPS data are available on request from REO, reobrien@wm.edu.

**Acknowledgement**

This work was supported by National Oceanic and Atmospheric Administration Grants No. NA13OAR4310072 and NA140AR4310132. KJB acknowledges support from the National Science Foundation. SHB and JDS acknowledges support from the US Environmental Protection Agency Award No. 835404, Electric Power Research Institute (EPRI), and National Oceanic and Atmospheric Administration Grant No. NA13OAR4310064. Special thanks to Dr. David Karl and Mr. Eric Grabowski, University of Hawaii, for the CHNS elemental analysis of DOM. DJR acknowledges support from the Gordan and Betty Moore Foundation award 6000 and the Simons Foundation SCOPE award 329108.

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



**Figures and Tables**
Table 1. Elemental ratios measured by SVN-AMS vs. other techniques for the various mixtures examined in this work.

| Sample | | O:C | H:C | N:C |
|---|---|---|---|---|
| Citric acid | Atomizer-AMS | 1.0 | 1.4 | -- |
| | SVN-AMS | 1.1 | 1.3 | -- |
| α-pinene SOA | Online-AMS | 0.48 | 1.6 | < 0.002 |
| | SVN-AMS | 0.50 | 1.6 | < 0.002 |
| Look Rock | Online-ACSM[a] | 0.13 ($f_{44}$=0.19) | 1.3 ($f_{43}$=0.062) | --[b] |
| | SVN-AMS[a] | 0.13 ($f_{44}$=0.16) | 1.3 ($f_{43}$=0.051) | --[b] |
| DOM | CHNS analyzer | ND | 1.74 | 0.080 |
| | SVN-AMS | 0.77 | 1.7 | 0.081 |


a.  Elemental ratios are estimated from parameterizations for $f_{44}$ and $f_{43}$ (Aiken et al., 2008; Ng et al., 2011b).
b.  There is no parameterization for N/C from UMR data.






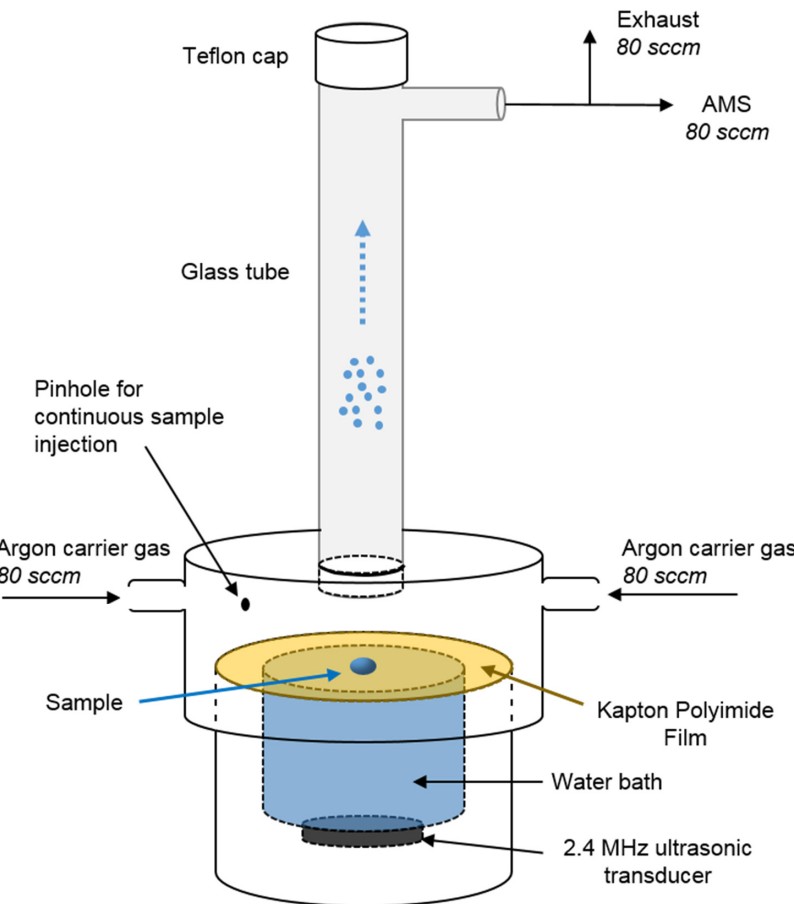

**Figure 1. Schematic diagram of small volume ultrasonic nebulizer (SVN). Samples (2-4µL) are loaded on the Kapton (or Teflon) film through either the hole in which the glass tube is seated (for discrete injections) or through the pinhole (for continuous injections). After the transducer is turned on, the aerosol is carried up through the glass tube and into the instrument by a ~160 sccm flow of house air or argon carrier gas. The water bath between the transducer and the Kapton film carries ultrasonic waves up to the film and serves to cool the ultrasonic transducer.**



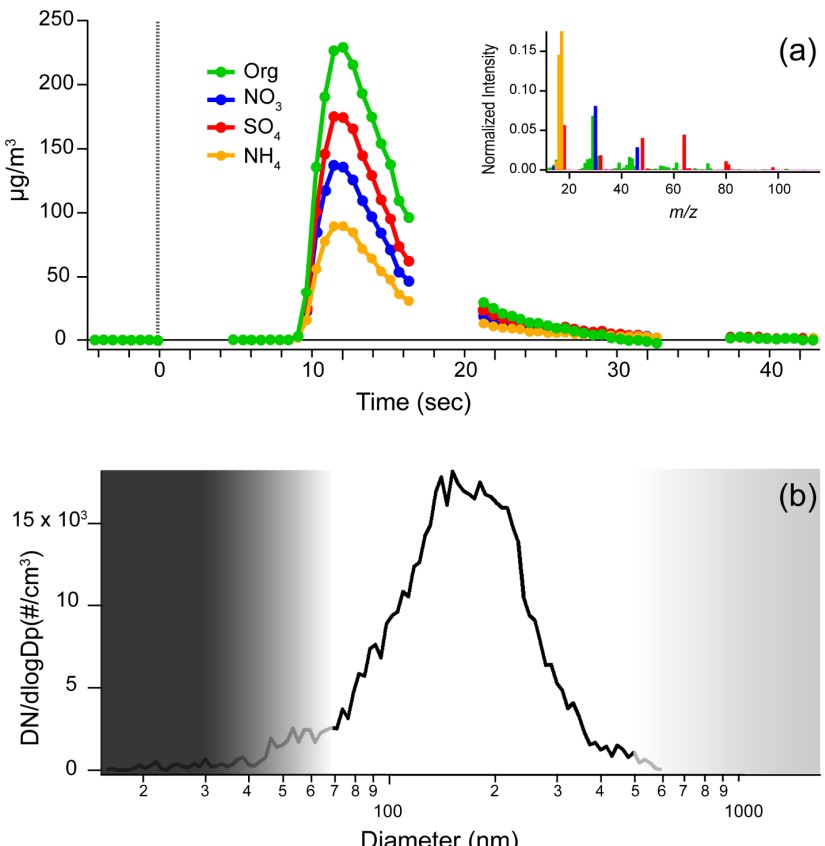

Figure 2. Measurements of the composition and size of nebulized samples from the SVN. (a) Time series of aerosol composition from a single 4 μL nebulization of an aqueous solution (mannitol, ammonium nitrate, and ammonium sulfate). Data were recorded using fast-mode MS for the AMS-open scans, with a mass spectrum collected every 0.5 s (filled circles). The gaps in the trace correspond to closed cycles where the aerosol beam was blocked to provide a background subtraction (gas-phase and instrument background) that was applied during data processing. Measured concentrations are not corrected for collection efficiency (CE) in the AMS, which affects the absolute values but not the relative concentrations. The inset shows the average mass spectrum acquired across the injection, normalized to total ion signal. (b) Aerosol size distribution from a ~ 1g/L citric acid solution measured with an SMPS (black line). The gradient represents the transmission efficiency for particles into the AMS with nearly 100% between 70-500 nm and decreased but substantial transmission for spherical particles 30-70 nm and 500 nm to 2.5 μm (Jimenez et al., 2003); thus, the smallest particles in the distribution will not be efficiently detected by the AMS.





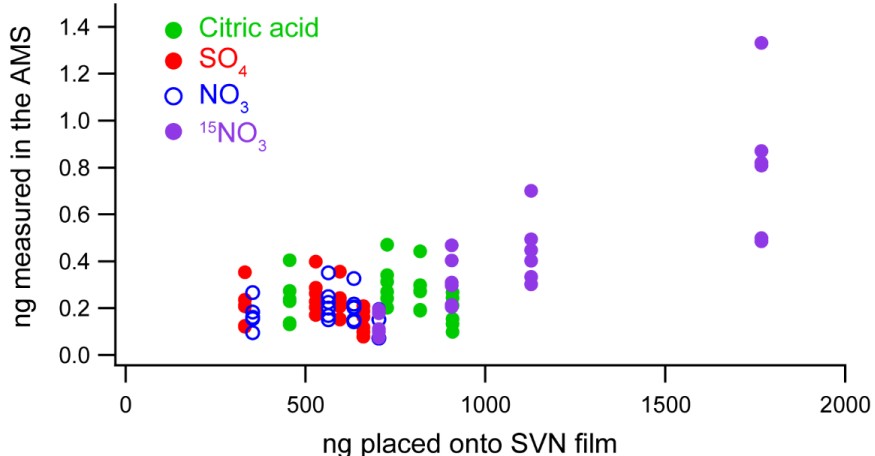

**Figure 3. Mass of each component placed on the thin film vs. the mass measured by the AMS for 4 different solutions with varying concentrations of citric acid, ammonium sulfate, ammonium nitrate, and the internal standard (NH₄¹⁵NO₃), all with a total solution concentration of 0.75 g/L. Each sample had 5 replicate injections, with the vertical spread in the measured masses indicating substantial run-to-run variability (up to a factor of 3) between injections.**





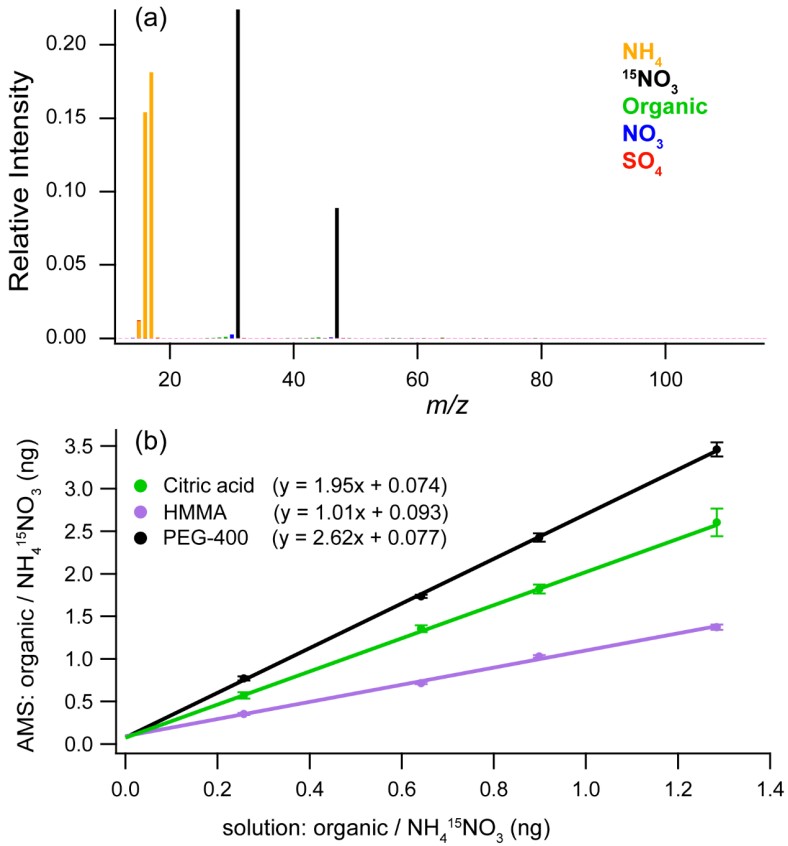

**Figure 4. (a) Blank of the Kapton film using 1 g/L internal standard solution ($^{15}$N- ammonium nitrate). (b) Calibration curves made using an internal standard for solutions with three different organic compounds: citric acid, 4-hydroxy-3-methoxy-DL-mandelic acid (HMMA), and polyethylene glycol 400 (PEG-400). The error bars are ±1σ for five replicate injections.**



607

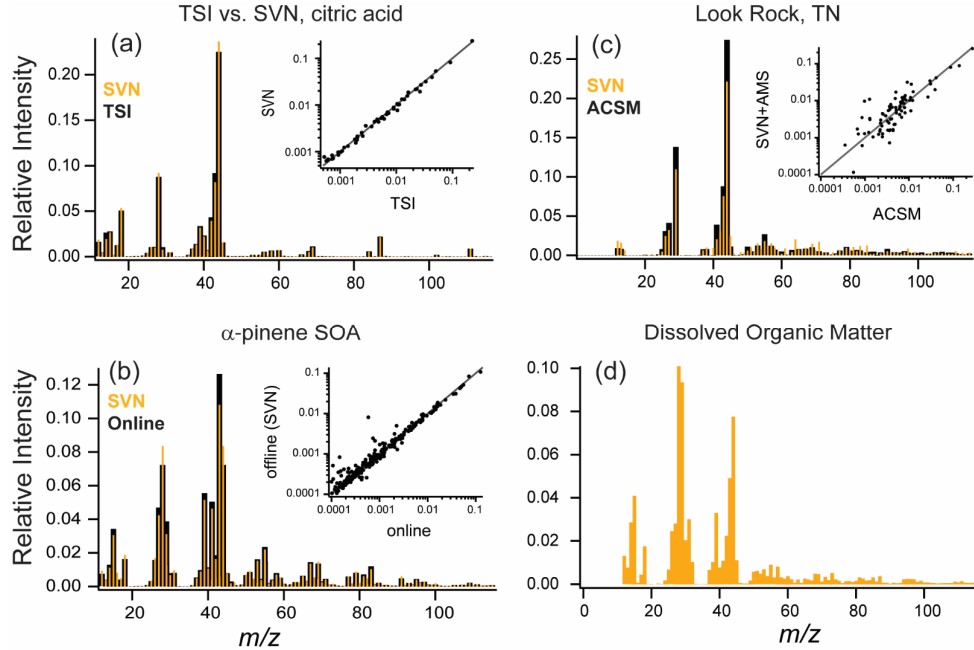

**Figure 5.** Online (or TSI atomizer) (black) vs. SVN nebulizer (orange) mass spectra for **(a)** an aqueous solution of citric acid at 1 g/L; **(b)** α-pinene + O₃ chamber SOA; **(c)** a SOAS campaign sample from Look Rock, TN with online data collected on an ACSM. Smaller insets in a, b, and c show direct comparison of intensities for each mass spectrum on a log scale. **(d)** AMS mass spectra from North Pacific Ocean dissolved organic matter nebulized with the SVN (since this sample was not from aerosol particles, no online samples are available).