# Peer review of "Ultrasonic Nebulization for the Elemental Analysis of Microgram-Level Samples with Offline Aerosol Mass"

_Atmospheric Measurement Techniques, 2018_

## Referee Comment (RC1) · Anonymous Referee #3 · 20 Nov 2018

Review of amt-2018-326 „Ultrasonic Nebulization for the Elemental Analysis of Microgram Level Samples with Offline Aerosol Mass Spectrometry"

General comments
In the manuscript by O'Brien et al. "Ultrasonic Nebulization for the Elemental Analysis of Microgram Level Samples with Offline Aerosol Mass Spectrometry" a novel analysis method combining aerosol generation with an ultrasonic nebulizer and an Aerosol Mass Spectrometer as a detector is presented. This work demonstrates the potential of the AMS to determine elemental composition of microgram-level of filter extracts or liquid samples. The manuscript describes the calibration process and investigates the effectiveness of this method for different samples of known and unknown composition. It fits in the scope of AMT and I would recommend it for publication after addressing specific comments listed below.

Specific comments
In general I find that there is some lack of information on the preparation and exact composition of the solutions used to test the effectiveness of the nebulization process (e.g. line 238/239 only a range is provided) and to determine the calibration curves (Fig. 4b e.g. what are the ratios of organic to $NH_4^{15}NO_3$?). The paper could be improved by providing more detailed information on the solutions. E.g. in Fig. 4b one cannot access what determines the ratio in the solution of organics to $NH_4^{15}NO_3$. With increasing mass is only the concentration of organics increased in the solution and $NH_4^{15}NO_3$ concentration is kept constant? Or the other way around? Are both concentrations varied for the different points on the calibration curve? Please provide more details. Also the reader would benefit much more if e.g. tables with the exact details of the used solution/mixtures where provided in the supplement.

In the manuscript the background signal of solvents (Milli-Q water and methanol) is mentioned several times but no graph or numbers are provided. It would be informative if e.g. in the supplemental material a graph could be shown to give the reader an estimate how much a background signal could contribute for both Milli-Q water and methanol to the actual signal of the sample.
line 54: it would be informative to give an example/a number of what "high temporal variability" means
line 92, line 138 and caption of Figure 1: What do you mean with "house air"? Please explain.
line 169: what does "low-NOx conditions". Please be more specific about the range of NOx concentrations during the experiments
Line 240 and Figure 3: "The amount of mass measured in the AMS increases slowly compared to the amount placed on the film…" This cannot be clearly seen from the graph because there is no information which combination of signals of citric acid, $NO_3$, $SO_4$ and $NH_4^{15}NO_3$ belong to one solution which was nebulized at the same time. According to the text the relative composition was changed for different samples only the sum of the solved components was kept constant. To better access a trend or the lack thereof it would be helpful if the reader could identify clearly the different samples in Figure 3.
Figure 3: $NH_4^{15}NO_3$ seem to show a somewhat linear response or at least a trend the other components are missing. This is not discussed in the text. Is this possibly due to the higher concentrations of $NH_4^{15}NO_3$ compared to the other components in the solution? Why was for $NH_4^{15}NO_3$ a higher concentration used than for the citric acid, $NH_4SO_4$, $NH_4NO_3$? Additionally it would be much more informative if for the y-axis error bars or at least some estimates where provided to judge better how significant the variability within the measurement error is.
Line 290ff: Compared are e.g. ratios of the signal of organic to the signal of $NH_4^{15}NO_3$ in the AMS to known ratios of organic to $NH_4^{15}NO_3$ in the solution. To correct for the variability due to the nebulization process a known amount of $NH_4^{15}NO_3$ is added to the sample. However e.g. if the composition of the sample is unknown the ratio of organics to the added $NH_4^{15}NO_3$ is also unknown. It is not clear to me since only the ratios of e.g. organics to $NH_4^{15}NO_3$ is used on the calibrations

curves how robust this method actually is if the ratio of organics to $NH_4^{15}NO_3$ is significantly different between what was used for the calibration curve and an unknown sample. The response might be different for different ratios of organics to $NH_4^{15}NO_3$. Unfortunately from the calibration curve it is not possible to access how the ratio on the x-axis for the known solution is composed. Was only $NH_4^{15}NO_3$ varied or only the organics or both?

Line 376: How does the internal standard improves CE of the AMS? If I am not mistaken that was not discussed in any of the previous sections of the paper. Please explain.

Technical comments

line 52: please explain once the abbreviation (CHNS) since not everyone necessary know what it stands for.

Figure S1: "((a) 2 g/L; size distribution centered at 200-300 nm) or more dilute solutions ((b) 0.2 g/L; size distribution centered at 100-200 nm)" Judging from the x-axis the maximum of the curves in both graphs seem to be centered around higher values. Please explain or correct.

Figure S1, S2: please add on both graph legends for the different traces shown

Line 307: Section number should be 3.3 instead of 3.1

Line 356: it is Figure 5d instead of 3d

---

## Referee Comment (RC2) · Anonymous Referee #2 · 26 Nov 2018

This manuscript proposes a new method to determine elemental ratios of microgram-level samples using offline AMS technique. Such technique would be quite useful and valuable, and therefore the paper merits publication. The description, justification and discussion of the technique is overall solid, this reviewer has a few comments for the authors to consider before its publication: (1) The manuscript aims to do elemental analysis, but as shown in the paper, it seems like you can also do mass quantification by using an internal standard. So why only mention elemental analysis? (2) Does the size distribution influence the measured particle composition? Also, for different samples, did you observe different size distributions? (3) Dehumidification is not applied in current experiments (although it can be done as you mentioned), therefore there might

be extra H2O signals influencing quantification of organics? I think you should add dehumidification procedure. (4) You mentioned there might be significant background signals if organic solvent is used to extract the samples. Did you try to use activated carbon to remove organic solvent? (5) You mentioned the ultrasonic nebulization may increase the temperature of your sample solution. This may lead to evaporation of some organics and therefore the composition and elemental ratios of your analysis. How to avoid this and how to consider such uncertainty? (6) Regarding the comparison of AMS mass spectra determined by SVN and online data, you need to be careful that the difference can attribute to a couple of factors: online measurement is for PM1 and can measure both water-soluble and water-insoluble species, while the SVN only determine water-soluble portion and your samples are PM2.5? (7) Why not use Canagaratna 2015 (Atmos. Chem. Phys. 2015, 15, (1), 253-272.) method to calculate H/C and O/C? Other typos: Line 307 3.1 mass spectral analysis. It is not 3.1 Line 316 atomizer (black) and the SVN (green), the colors are inconsistent Line 323èąŇ offline (red) vs. online (black)ïijŇthe colors are inconsistent

---

## Referee Comment (RC3) · Anonymous Referee #1 · 28 Nov 2018

This is an interesting paper by developing a small volume nebulizer for elemental analysis with aerosol mass spectrometer. The major advantage of this technique is the volume of samples needed for analysis. This manuscript is generally well written, and I recommend it for publication after addressing the following comments.

1. The future applications of this technique can be expanded, particularly compared with previous AMS offline analysis. In general, the volume of DI-water extracted solutions from filter samples collected with high-volume samplers are not an issue for elemental analysis with AMS. Then why we need such a technique for offline AMS analysis?

[Figure]

2. The authors didn't show any high resolution mass spectra of compounds or samples analyzed in this study. For example, North Pacific Ocean sample in Figure 5d. Clear signals of m/z 78 ($CH_2SO_2^+$) and 79 ($CH_3SO_2^+$) are expected, which were not. Another question is the minimum concentration used for the SVN-AMS analysis. Because "fast MS" mode was used for discrete samples, signal-to-noise ratio could can be an issue for high resolution peak fitting.

3. It is not recommended to directly compare the mass spectra between ACSM and AMS. ACSM often presents much higher m/z 44 than AMS [Fröhlich et al., 2015], and O/C estimated with f44 can also have a large uncertainty.

4. Typos of "FIGERO-CIMS"(line 82) and "Figure 3d" (line 356).

Fröhlich, R., et al. (2015), ACTRIS ACSM intercomparison – Part 2: Intercomparison of ME-2 organic source apportionment results from 15 individual, co-located aerosol mass spectrometers, Atmos. Meas. Tech., 8(6), 2555-2576, doi:10.5194/amt-8-2555-2015.

---

## Referee Comment (RC4) · Anonymous Referee #4 · 5 Dec 2018

The work presented by O'Brien et al. tested a new offline method (an ultrasonic neb-ulizer combined with an AMS) for detecting organic matter in environmental samples. This method is of interest because it requires only small sample volume. However, the advantage of this technique is overlooked in the paper. For example, the authors need to consider the preparation process of the sample liquid as I commented below. The samples tested in this study are very limited. Thus the conclusions need careful modifications to avoid misleading. Moreover, the description about the sample prepa-rations need substantial improvements. The internal standard method is not yet fully applicable, which could be tested in real ambient samples. The pretreatments for DOM samples seems intensive. Whether if it is necessary for SVN-AMS is unclear. I would

suggest to use the SVN-AMS method directly for water samples for DOM and the effects of pretreatment steps should be investigated. Overall, this manuscript needs a major revision before publishing on AMT.

Specific comments:

Line 69-74 and Line 83-84: While the analysis of SVN-AMS uses only a few $\mu$L (0.4 $\mu$g solute), preparing the sample liquids require additional volume. For examples, for ambient aerosol particles, it perhaps still takes at least a few mL to dissolve the material on filters into a solution. Then, to achieve the advantage of less required material, the SVN-AMS method needs substantial preconcentration (e.g., drying by purging N2 as described in Line 180-188). If preconcentration is applicable for organic aerosol, the atomizer-AMS method can go with it too. The real difference of this method compared to the atomizer-AMS method seems the acceptable preconcentration magnitude, meaning the sample material into $\mu$L can be much more concentrated than into mL liquid. But intensive concentration may cause artifacts and only work for some environmental samples. Please clarify and avoid misleading.

Line 139-140: This study did not use Argon, right? Please clarify.

Line 152-155, Line 209-215, and Figure 2 caption: I am confused about the description of particle size and necessary sample concentration. This study did not use a dryer (Line 104-105). Was SMPS data for dried or wet particles? Why the diameters of dried nebulized particles is important for AMS transmission? My understanding of this approach is that the AMS only samples whatever wet aerosols were partially dried in aerodynamic lens and went through.

Figure S1. The SVN-AMS uses fast MS mode which is different from PTOF. Why to use PTOF data to determine the minimum sample concentration. The authors need to identify the minimum sample concentration for AMS detection at normal operation conditions (i.e., the integrated mass compared to the detection limit and the background levels).

Line 170-172: Was the chamber run in a batch or continuous mode? Is the seed concentration of 60 $\mu$g/m3 the initial concentration in the chamber? What about the 500 ppbv ozone? Is that initial concentration as well? Do the concentrations vary over time? The filter samples were collected for 10 hr. How did the particle concentration vary over that 10 hr period? In Figure 5b, is the online AMS spectra averaged for that 10 hours?

Line 172-173, Line 178-179, and Line 324: It is not clear to me how the 10-min sampling can represent the "blank". What kind of blank? Before the start of the experiments, what do the filters collect? Can 10 min be enough? What do the authors mean "blank subtraction was carried out with a scaling of the filter blank to 12% of the sample signal, as determined from the internal standard in each sample"? The internal standard is not mentioned in the previous text.

Line 180-186: Although the filter samples are from another study, the authors should provide enough details about the sample/samples used in this study. What is the sampling period? "Blown down to dryness" means completely dry? I suspect completely dryness affects the semivolatile SOA species. If not, how concentrated the solution is used for SVN-AMS measurements compared to ambient loading.

Line 190-191: The grade of the reagents should be provided.

Line 198-199: How is this white power prepared for the SVN-AMS analysis? Dissolve into additional water or just melt?

Line 200: Please clarify for each case (chamber, SOAS, and DOM), how many fiters/samples were used in this study.

Line 240: The description about the variations should be more quantitative. Given such big variations, I think it is necessary to clearly state that this method is not good for quantitative analysis of the mass concentrations of the samples.

Line 280-281: Where are the background coming from? If adding the internal standard

to a sample solution, what happens?

Line 293-306: The internal standard method should be tested for ambient samples with the suggested RIE of 1.4; otherwise the goodness of this method remains unclear.

Section 3.3: The run-to-run variations is large for nebulization efficiency. What about the run-to-run variations of elemental composition and mass spectra. Please provide.

Line 316: The model of the TSI atomizer as well as the operating conditions should be provided.

Line 320: I don't understand how the dot product (of what?) represents the similarity of the two spectra.

Line 344-353: The mass spectra showed in Figure 5c are indeed quite different in terms of many distinguished ions. I think saying the two mass spectra have a high degree of agreement (Line 344) or one is well represented by the other (Line 351-352) is improper and misleading. Since the major similarity appears for m/z 41-44, the authors may present some of the major peak ratios (e.g., f44-to-f43) instead. Also, in Table 1, the uncertainty of elemental ratios should be provided. Using f44 to derive O:C is associated with much greater uncertainty than the AMS method does. The comparison for Look Rock somewhat indicates that the elemental ratios are less sensitive to the method.

Line 355-356: Not only "remain in the condensed phase after nebulization" but also after intensive pretreatments (e.g., serial dilution/concentration etc. in Line 197-198).

Line 365-366: Why the pretreatments in Line 197-198 are needed for the SVN-AMS method for DOM? Tests on real water samples should be presented. The effects of dilution or concentration on the analysis deserve further discussions.

Technical Remarks: Line 27: What kind of "diameter"? Also, to be clear, this diameter is for dry particles. Line 29 and Line 281: Please add "material" after "organic". Figure 2: Please properly write the ions in the legend. Line 240: "Slowly" is an improper word

here, maybe "slightly". Figure 5: To be clear, the "SVN" in panels b), c), d) is indeed "SVN-AMS". In panel a), the legend of "TSI" should be "Atomizer". And the "Online" in panel c) should be "AMS"? Line 307: "3.1" should be "3.3".

---

## Author Comment (AC1) · 14 Jan 2019

Review of amt-2018-326 "Ultrasonic Nebulization for the Elemental Analysis of Microgram Level Samples with Offline Aerosol Mass Spectrometry"

**General comments**

**In the manuscript by O'Brien et al. "Ultrasonic Nebulization for the Elemental Analysis of Microgram Level Samples with Offline Aerosol Mass Spectrometry" a novel analysis method combining aerosol generation with an ultrasonic nebulizer and an Aerosol Mass Spectrometer as a detector is presented. This work demonstrates the potential of the AMS to determine elemental composition of microgram-level of filter extracts or liquid samples. The manuscript describes the calibration process and investigates the effectiveness of this method for different samples of known and unknown composition. It fits in the scope of AMT and I would recommend it for publication after addressing specific comments listed below.**

We thank the reviewer for their helpful comments and suggestions. We have added text to the manuscript clarifying questions and comments raised by the reviewer.

**Specific comments**

**In general I find that there is some lack of information on the preparation and exact composition of the solutions used to test the effectiveness of the nebulization process (e.g. line 238/239 only a range is provided) and to determine the calibration curves (Fig. 4b e.g. what are the ratios of organic to NH4 15NO3?). The paper could be improved by providing more detailed information on the solutions. E.g. in Fig. 4b one cannot access what determines the ratio in the solution of organics to NH4 15NO3. With increasing mass is only the concentration of organics increased in the solution and NH4 15NO3 concentration is kept constant? Or the other way around? Are both concentrations varied for the different points on the calibration curve? Please provide more details. Also the reader would benefit much more if e.g. tables with the exact details of the used solution/mixtures where provided in the supplement.**

We thank the reviewer for this suggestion and have added a table showing the concentrations used in the samples that generated figures 3 and 4 in the supplemental. We have also added text to the manuscript directing the reader there for further information.

**In the manuscript the background signal of solvents (Milli-Q water and methanol) is mentioned several times but no graph or numbers are provided. It would be informative if e.g. in the supplemental material a graph could be shown to give the reader an estimate how much a background signal could contribute for both Milli-Q water and methanol to the actual signal of the sample.**

When we atomize pure solvents, we observe no signal in the AMS because the concentration of trace components/contaminants is too small to generate aerosols of a large enough dimeter to pass through the aerodynamic lens and reach the vaporizer of the AMS. We address this in the second paragraph in section 3.1. To show the very low background observed for Milli-Q samples run with sufficient analyte to generate aerosols measureable in the AMS, we provide Figure 4a. To clarify this point we have added

the following text to the end of the second paragraph in section 3.2.2: "*For all tests of background signals and blanks, the internal standard is added to the solutions at concentrations between 0.5-1 g/L in order generate aerosols of sufficient size for the AMS.*"

Analysis of the background signal from methanol and other organic solvents, when sufficient analyte is present in the solution to generate aerosols, is an area of active research for the first author as mentioned in the conclusions.

**line 54: it would be informative to give an example/a number of what "high temporal variability" means**

We have added text that provides the example of air masses in major urban regions being a system with a relatively rapidly varying aerosol composition.

**line 92, line 138 and caption of Figure 1: What do you mean with "house air"? Please explain. line 169: what does "low-NOx conditions". Please be more specific about the range of NOx concentrations during the experiments**

House air is zero air from an Aadco zero air generator. This information has been added to the beginning of the experimental and the figure caption has been changed to "zero air". In the chamber $NO_x$ was less than 10 ppb. This information has been added to the text.

**Line 240 and Figure 3: "The amount of mass measured in the AMS increases slowly compared to the amount placed on the film..." This cannot be clearly seen from the graph because there is no information which combination of signals of citric acid, NO3, SO4 and NH4 15NO3 belong to one solution which was nebulized at the same time. According to the text the relative composition was changed for different samples only the sum of the solved components was kept constant. To better access a trend or the lack thereof it would be helpful if the reader could identify clearly the different samples in Figure 3.**

The samples used to generate Figure 3 are the sample solutions run for the citric acid calibration curve in Figure 4. For the efficiency analysis, 4 μL of each solution was placed on the kapton film and atomized with six replicates run for each sample. Information on the samples used and their corresponding locations in Figure 3 has been added to the supplemental and text directing the reader to that information has been added to the manuscript text.

**Figure 3: NH4 15NO3 seem to show a somewhat linear response or at least a trend the other components are missing. This is not discussed in the text. Is this possibly due to the higher concentrations of NH4 15NO3 compared to the other components in the solution? Why was for NH4 15NO3 a higher concentration used than for the citric acid, NH4SO4, NH4NO3? Additionally it would be much more informative if for the y-axis error bars or at least some estimates where provided to judge better how significant the variability within the measurement error is.**

We thank the reviewer for this comment and have added material to both the supplemental as well as the manuscript to clarify this topic.

We have added detailed information on the solutions used to generate Figure 3 to the supplemental material. A higher concentration of the labeled ammonium nitrate was used because it is the internal standard and the concentrations of the other components are varied relative to it. The vertical column of data points are 6 replicate injections of the same solution and are shown to provide a measure of the variability. The relative amounts of total signal observed for any given sample can vary, and we find that the trend shown here is not always observed. Thus, the trend the reviewer observes is not inherent to the measurement but was observed for this sample. What is consistent across all measurements is the efficiencies on the order of 0.02-0.06% and the ratios between the internal standards and the analyte being proportional to the solutions.

We have modified this section of the text to clarify this:

"Six replicate injections of 4 $\mu$L drops of the solutions from one of the calibration curves (section 3.2.2 below) were atomized, and the total mass observed in the AMS was calculated as described above. (Details on the concentrations of analytes in these calibration solutions for Figures 3 and 4 are provided in the supplemental.) There are variations in the efficiency from sample to sample and run to run, thus the trends shown in Figure 3 are illustrative only. The key trait observed is that the measured nebulization efficiencies are on the order of 0.02-0.06%, indicating that the aerosol mass detected with the AMS is approximately three orders of magnitude lower than the mass originally deposited on the thin film."

**Line 290ff: Compared are e.g. ratios of the signal of organic to the signal of NH4 15NO3 in the AMS to known ratios of organic to NH4 15NO3 in the solution. To correct for the variability due to the nebulization process a known amount of NH4 15NO3 is added to the sample. However e.g. if the composition of the sample is unknown the ratio of organics to the added NH4 15NO3 is also unknown. It is not clear to me since only the ratios of e.g. organics to NH4 15NO3 is used on the calibrations curves how robust this method actually is if the ratio of organics to NH4 15NO3 is significantly different between what was used for the calibration curve and an unknown sample. The response might be different for different ratios of organics to NH4 15NO3 . Unfortunately from the calibration curve it is not possible to access how the ratio on the x-axis for the known solution is composed. Was only NH4 15NO3 varied or only the organics or both?**

This question raises an important point about how to implement the use of internal standards for quantification. When the concentration of the analyte is unknown in a sample, initial tests must be run to verify the range of concentrations. Then, an appropriate amount of internal standard can be added such that the ratio of analyte to internal standard matches the range used in the calibration solutions. If this is not possible, possibly due to sample mass limitations, the calibration curve can be subsequently remade to encompass the observed approximate concentrations. For the solutions run here, the IS standard was kept the same and the analyte concentrations were varied.

The concentrations for the solutions used here have been added to the supplemental and we have added text to the third paragraph in section 3.2.2 clarifying this for the reader.

"For quantification of unknowns, known concentrations of the internal standard are added to the samples *at ratios comparable to what is used for the calibration curve.* The ratio of the measured AMS signals can then be used to calculate the unknown analyte concentration from the calibration curve."

**Line 376: How does the internal standard improves CE of the AMS? If I am not mistaken that was not discussed in any of the previous sections of the paper. Please explain.**

The use of an organic internal standard may improve collection efficiency as it may reduce particle bounce off the vaporizer in the AMS.  This has been added to the sentence in the conclusions.

**Technical comments**

**line 52: please explain once the abbreviation (CHNS) since not everyone necessary know what it stands for.**

We have added "carbon, hydrogen, nitrogen, and sulfur" in front of CHNS.

**Figure S1: "((a) 2 g/L; size distribution centered at 200-300 nm) or more dilute solutions ((b) 0.2 g/L; size distribution centered at 100-200 nm)" Judging from the x-axis the maximum of the curves in both graphs seem to be centered around higher values. Please explain or correct.**

This has been corrected.

**Figure S1, S2: please add on both graph legends for the different traces shown**

These have been added

**Line 307: Section number should be 3.3 instead of 3.1**

This has been corrected

**Line 356: it is Figure 5d instead of 3d**

This has been corrected

---

## Author Comment (AC2) · 14 Jan 2019

**This manuscript proposes a new method to determine elemental ratios of microgram level samples using offline AMS technique. Such technique would be quite useful and valuable, and therefore the paper merits publication. The description, justification and discussion of the technique is overall solid, this reviewer has a few comments for the authors to consider before its publication:**

We thank the reviewer for their comments/questions/and suggestions and we have made changes to the manuscript to address their concerns as detailed below.

**(1) The manuscript aims to do elemental analysis, but as shown in the paper, it seems like you can also do mass quantification by using an internal standard. So why only mention elemental analysis?**

We are very interested in the quantification capabilities of the technique. The discussion on quantification in section 3.2 and figure 4 are laying the groundwork for this type of analysis. We have added a note in the conclusions that the SVN will be used to generate aerosol for quantitative and qualitative analysis of environmental samples in the future.

**(2) Does the size distribution influence the measured particle composition? Also, for different samples, did you observe different size distributions?**

The current model of the SVN is better suited for discrete samples and particle size measurements tend to require a continuous source of aerosol for at least a minute or two. Initial tests were carried out using continuous flow in the SVN and these results are shown in Figures 2 and S1. Figure S1 shows that lower concentrations make smaller sized particles. It also shows that we observe homogenous particles indicating that the size range sampled will not vary the composition measured in the AMS.

We have added a sentence to the caption on Figure S1 to highlight this: "For these samples, the size distribution of the components is fairly uniform consistent with the formation of homogenous particles in the nebulizer."

**(3) Dehumidification is not applied in current experiments (although it can be done as you mentioned), therefore there might be extra H2O signals influencing quantification of organics? I think you should add dehumidification procedure.**

We agree with the reviewer that care with quantification is very important. The AMS software used to process these data sets limits the organic $H_2O$ signal to 0.225 of the $CO_2^+$ signal measured in the sample to account for the presence of water in the particles. The addition of a dehumidication procedure may provide valuable insights for some studies which is why it is raised in the paper. However, we caution against the assumption that after the particles have passed through a dehumidifier all the water observed in the AMS is due to organic pyrolysis/fragmentation as this may not be true across all sample types.

**(4) You mentioned there might be significant background signals if organic solvent is used to extract the samples. Did you try to use activated carbon to remove organic solvent?**

We thank the reviewer for this suggestion. The lead author is currently working on characterizing background signals from organic solvents and is excited to test this idea out as a denuder before sampling into the AMS. This method will likely improve the removal of organic solvent from the particles, which will help characterize lower-volatility organic contaminants present in the solvent.

**(5) You mentioned the ultrasonic nebulization may increase the temperature of your sample solution. This may lead to evaporation of some organics and therefore the composition and elemental ratios of your analysis. How to avoid this and how to consider such uncertainty?**

We have begun the characterization of this feature by the offline vs. online comparisons. Our initial tests, shown in Figure 5a and 5b show very good reproducibility between the chemical composition measured in the AMS after SVN nebulization and what has not been ultrasonically nebulized. To further reinforce this we have added text to the conclusions (italic = additional text): "A direct comparison between the mass spectra generated by commercial spray atomizers or by real-time aerosol particles sampled directly from the atmosphere showed high degrees of agreement*, indicating minimal composition changes during nebulization*."

**(6) Regarding the comparison of AMS mass spectra determined by SVN and online data, you need to be careful that the difference can attribute to a couple of factors: online measurement is for PM1 and can measure both water-soluble and water-insoluble species, while the SVN only determine water-soluble portion and your samples are PM2.5?**

We agree that the differences in size distribution and the effects of solubility may influence the composition observed. We address this for both the chamber and the ambient comparisons in section 3.3 and have added text highlighting the size differences for the chamber experiment between what the measured online (PM1) and what is collected on the filters (no cut-off was applied).

**(7) Why not use Canagaratna 2015 (Atmos. Chem. Phys. 2015, 15, (1), 253-272.) method to calculate H/C and O/C?**

H/C and O/C values for the HR-ToF-AMS data sets were calculated using the above reference. We have clarified this in the manuscript by adding (new text italics): "For samples compared to chamber or ambient online-AMS data sets, house air was the carrier gas, standard empirical estimates were used, *and the improved-ambient method for elemental ratios was applied* (Canagaratna et al., 2015)."

**Other typos: Line 307 3.1 mass spectral analysis. It is not 3.1**

This has been corrected

**Line 316 atomizer (black) and the SVN (green), the colors are inconsistent Line 323 offline (red) vs. online (black) the colors are inconsistent**

This has been corrected

---

## Author Comment (AC3) · 14 Jan 2019

**This is an interesting paper by developing a small volume nebulizer for elemental analysis with aerosol mass spectrometer. The major advantage of this technique is the volume of samples needed for analysis. This manuscript is generally well written, and I recommend it for publication after addressing the following comments.**

We thank the reviewer for their helpful comments and suggestions. We have added text to the manuscript to address the questions and comments raised.

**1. The future applications of this technique can be expanded, particularly compared with previous AMS offline analysis. In general, the volume of DI-water extracted solutions from filter samples collected with high-volume samplers are not an issue for elemental analysis with AMS. Then why we need such a technique for offline AMS analysis?**

The smaller droplet sizes generated in the nebulizer, compared to commercial atomizers minimizes the influence of background material from the solvent. For a single injection of an aqueous solution, the SVN-AMS requires only 400 ng of material and preliminary work on organic solvents shows an even better nebulization efficiency. This small size enables the analysis of trace samples as, for example, samples collected with impactors over short periods of time. Additionally, the SVN-AMS is a platform that enables direct comparison of samples prepared for other offline analyses, including both liquid chromatography and direct electrospray ionization of samples into mass spectrometers, as the concentrations needed for both analyses are similar. Finally, this platform decouples the gas flow rate from the atomization process enabling a concentration of the aerosol packet, if needed.

All of these advantages are mentioned in the manuscript except for the comparison with soft ionization techniques and the decoupling of the gas flow rate with the nebulization process. Text has been added to the end of the introduction and the conclusions:

*"The concentration ranges needed (described below) are comparable to the concentrations used for other offline characterizations including soft ionization with electrospray ionization into mass spectrometers. Thus, this technique provides a platform for direct comparison between offline-AMS samples and other analytical techniques."*

*"Finally, in contrast to atomizers (where the carrier gas generates the aerosol), ultrasonic nebulizers decouple the aerosol formation from the carrier gas flow rate, enabling potential concentration of the aerosol prior to sampling."*

**2. The authors didn't show any high resolution mass spectra of compounds or samples analyzed in this study. For example, North Pacific Ocean sample in Figure 5d. Clear signals of m/z 78 (CH2SO2+) and 79 (CH3SO2+) are expected, which were not. Another question is the minimum concentration used for the SVN-AMS analysis. Because "fast MS" mode was used for discrete samples, signal-to-noise ratio could can be an issue for high resolution peak fitting.**

The DOM from the North Pacific Ocean is the high molecular weight fraction of the organic material and, as such, has had the lower molecular weight compounds (including any possible methane sulfonic acid) removed. We do not expect signals from methane sulfonic acid derivatives in our samples. The North

Pacific sample is not total DOM, but the polysaccharide fraction (~ 25% total DOM) isolated as described between lines 197 and 204.  To further clarify this point, we have explicitly changed the text in the abstract, experimental, and results and discussion to clarify that this is the polysaccharide fraction of DOM. We have also added text to the end of section 3.3 clarifying this:

*"Here we demonstrate the analysis of the high molecular weight fraction of the polysaccharide fraction of dissolved organic matter (DOM) with the SVN-AMS.  The DOM sample was prepared using a standard protocol for the isolation of this fraction of the organic material (see section 2.3).  This preparation removes the lower molecular weight compounds so chemicals such as methane sulfonic acid are not expected to be observed."*

We agree that signal to noise can be a concern for peak fitting.  For all analyses at least three replicate injections are carried out.  The average mass spectra across these injections is then used for the peak fitting.  If the observed signal is low during collection, more replicate injections are carried out to help improve the S/N during analysis.

We have added the following text to the experimental to clarify this:

"*For the high resolution  peak fitting and the analysis of the mass spectrum and the elemental ratios, the average mass spectrum across all injections is used.  For quantification, the total signal under each injection pulse (see below, Figure 2a) is used.*"

**3. It is not recommended to directly compare the mass spectra between ACSM and AMS. ACSM often presents much higher m/z 44 than AMS [Fröhlich et al., 2015], and O/C estimated with f44 can also have a large uncertainty.**

We agree that comparisons between ACSM and the AMS should be carried out very carefully and only present the comparison because no on-line AMS data was available for the Look Rock samples.  We have added text to the Results and Discussion section (3.3) highlighting and clarifying this:

"The high degree of overlap *in the intensities of the dominant ions* between the online (AMS/ACSM) measurements and offline (SVN-AMS) results indicates that the ensemble organic composition for these aerosol samples is generally well-represented by the SVN-AMS measurements (Table 1). *However, the estimated elemental ratios from a lower resolution AMS are more uncertain than from the HR-ToF-AMS. Thus, the  ratios for these samples in Table 1 are provided only as a demonstration of the overall agreement between the two techniques.*"

**4. Typos of "FIGERO-CIMS"(line 82) and "Figure 3d" (line 356).**

These have been corrected.

**Fröhlich, R., et al. (2015), ACTRIS ACSM intercomparison – Part 2: Intercomparison of ME-2 organic source apportionment results from 15 individual, co-located aerosol mass spectrometers, Atmos. Meas. Tech., 8(6), 2555-2576, doi:10.5194/amt-8-2555- 2015.**

---

## Author Comment (AC4) · 14 Jan 2019

**The work presented by O'Brien et al. tested a new offline method (an ultrasonic nebulizer combined with an AMS) for detecting organic matter in environmental samples. This method is of interest because it requires only small sample volume. However, the advantage of this technique is overlooked in the paper. For example, the authors need to consider the preparation process of the sample liquid as I commented below. The samples tested in this study are very limited. Thus the conclusions need careful modifications to avoid misleading. Moreover, the description about the sample preparations need substantial improvements. The internal standard method is not yet fully applicable, which could be tested in real ambient samples. The pretreatments for DOM samples seems intensive. Whether if it is necessary for SVN-AMS is unclear. I would suggest to use the SVN-AMS method directly for water samples for DOM and the effects of pretreatment steps should be investigated. Overall, this manuscript needs a major revision before publishing on AMT.**

We would like to thank the reviewer for their helpful and thorough comments. We have made the requested corrections and have used the authors comments, questions, and suggestions to clarify material covered in the manuscript. Specifically, we have added text that communicates the sample preparation processes that are necessary for some samples and the corresponding limitations/care that must be exercised.

The samples provided as examples in the text are demonstrations of the technique applied to different types of systems. Work is actively being carried out to expand these examples to other systems including the characterization of the internal standard method for ambient samples. The work communicated in this manuscript is an explanation of the method and a demonstration of its potential. Given the linearity of the response to the calibration curves with the internal standard, we believe that the applicability of the technique is only limited by the correct choice of standard to serve as the calibrant. Fortunately, this technique also provides a platform to test multiple different standards, enabling later adjustments as more knowledge of the system is gained via other analyses.

The information in the experimental on sample preparation has been expanded as requested. The pretreatments for DOM were necessary to investigate the elemental ratios of the organic mixture in the samples. Marine dissolved organic matter is a very complex mixture of organic compounds that together occur at low concentrations (~ 0.5-1 mg/L) within a very saline solution (~ 35 g/L). Given the very high ratio of salt to organic C and N (up to $1x10^6:1$), it is unlikely that direct measurement of elemental DOM ratios in seawater itself would be successful. There are reasonably sensitive techniques to measure total DOC, DON, and DOP in seawater. The AMS method described here is not offered as an alternative to these methods but does provide a platform for generating comparable data with decreased sample mass requirements.

Virtually all analytical and experimental work designed to study DOM composition and microbial cycling use some form of physical or chemical concentration of DOM from seawater. The ultrafiltration isolation technique used here for our DOM sample is commonly applied by marine chemists to study DOM cycling of this semi-labile fraction. Elemental analyses of this fraction are ill suited for high throughput experimental work. Indeed, we expect the AMS method might be a game changer in the ability to perform high throughput analyses of DOM elemental ratios in experimental studies. Since the same pre-treatments must be applied to the sample before either CHNS or SVN-AMS analysis, this data set provides a good test of the technique to characterize very complex organic mixtures.

We agree with the reviewer that the effects of pretreatment steps should be investigated and are excited for the idea to compare samples prepared with the method used for SVN (which is also often carried out to prepare samples for electrospray ionization/MS) to samples prepared and atomized with commercial atomizers. Bridging the gap between different techniques in terms of sample preparation is a challenging problem and the quantitative and qualitative capabilities of the AMS, when coupled to the SVN and other atomizers, may provide a powerful tool to characterize these effects.

**Specific comments:**

**Line 69-74 and Line 83-84: While the analysis of SVN-AMS uses only a few µL (0.4 µg solute), preparing the sample liquids require additional volume. For examples, for ambient aerosol particles, it perhaps still takes at least a few mL to dissolve the material on filters into a solution. Then, to achieve the advantage of less required material, the SVN-AMS method needs substantial preconcentration (e.g., drying by purging N2 as described in Line 180-188). If preconcentration is applicable for organic aerosol, the atomizer-AMS method can go with it too. The real difference of this method compared to the atomizer-AMS method seems the acceptable preconcentration magnitude, meaning the sample material into µL can be much more concentrated than into mL liquid. But intensive concentration may cause artifacts and only work for some environmental samples. Please clarify and avoid misleading.**

We thank the reviewer for this comment, the comparison between different atomization techniques is an important area of research as we characterize environmental samples with different offline techniques.

For most filter samples, the SVN requires pre-concentration. We have added text to the introduction highlighting this:

*"In some cases, depending on the sample, pre-concentration is required to generate suitable solutions for analysis. The concentration ranges needed (described below) are comparable to the concentrations used for other offline characterizations including soft ionization with electrospray ionization into mass spectrometers. Thus, this technique provides a platform for direct comparison between offline-AMS samples and other analytical techniques."*

We agree that care should always be a taken with sample handling. We have also added text to the conclusions highlighting this fact.

*"For these samples, pre-concentration was required to prepare a suitable solution concentration for analysis. This will be required for some types of environmental samples and care should be taken to minimize artifacts during solution preparation."*

Here, the SVN-AMS is not presented as a replacement for atomizer-AMS techniques and we agree that some atomizer-AMS techniques could be used to characterize samples prepared in the manner discussed in this manuscript. The SVN is presented as a new, additional, option for aerosol generation with strengths that will make it very suitable for the analysis of certain types of samples. The SVN requires very small sample volumes per injection (2-4 µL), it generates small aerosols thus reducing the effect of organic contamination from the solvent, and it enables increases in the aerosol particles per

unit volume by decoupling the aerosol formation process from the carrier gas flow rate. Text communicating these strengths has been added to the introduction and conclusion sections.

**Line 139-140: This study did not use Argon, right? Please clarify.**

This study used Argon for the DOM and for the citric acid comparison. As the chamber and ambient samples had air background, we used zero air for those samples. We have added the following text (additions in italics)

"With the SVN, inert carrier gases such as argon can also be used, allowing for the direct measurement of the $CO^+$ ion intensity (as demonstrated below for dissolved organic matter, *the majority of the other samples were run with zero air*)."

**Line 152-155, Line 209-215, and Figure 2 caption: I am confused about the description of particle size and necessary sample concentration. This study did not use a dryer (Line 104-105). Was SMPS data for dried or wet particles? Why the diameters of dried nebulized particles is important for AMS transmission? My understanding of this approach is that the AMS only samples whatever wet aerosols were partially dried in aerodynamic lens and went through.**

The particles that were measured in the SMPS were likely partially dried as the carrier gas was dry zero air. The minimum size necessary for sampling into the AMS is ~70 nm. If the particles dry during transit between the nebulizer and the AMS to diameters less than this, they will have very low transmission into the AMS.

If the solution that is being atomized is too dilute, the particles that are formed may be too small to pass into the AMS.

We have added text to the manuscript to clarify this (added text in italics):

"Assuming that the density of the dried particle is 1.3 g/cm$^3$ (Nakao et al., 2013), the minimum sample concentration that will form a 100 nm dried particle is approximately 0.3 g/L. *More dilute solutions do not generate signal in the AMS because the majority of the aerosol particles that are formed are too small for transmission through the aerodynamic lens of the AMS (Figure 2b)*. To generate large enough aerosol particles from more dilute solutions, larger initial droplets could be formed by changing the transducer *to one* that vibrates at a lower frequency."

**Figure S1. The SVN-AMS uses fast MS mode which is different from PTOF. Why to use PTOF data to determine the minimum sample concentration. The authors need to identify the minimum sample concentration for AMS detection at normal operation conditions (i.e., the integrated mass compared to the detection limit and the background levels).**

The PTOF data is not used to determine the minimum sample concentration. We have added text to the caption of Figure S1 to clarify this.

The total signal observed in the AMS was used to determine the minimum concentration needed.  The text added addressing the comment above also covers this idea.  The limitation for sample concentration comes dominantly from the maximum aerosol particle size that can be formed. Enough material is needed to form aerosols that can be sampled in the AMS. Beyond this, the limitations in the AMS roughly match what is observed with online work: measured particle concentrations below ~4 $\mu g/m^3$ have higher noise and are not ideal for chemical analysis. We find that concentrations greater than 0.2 g/L provide consistently good quantitative and qualitative results for organic samples. That is where the lower value for the range comes from (4 $\mu g$ of material).  For the lower concentrations, the addition of the internal standard actually benefits the results, as the overall sample concentration is higher.

We have added text to the manuscript to clarify how this can be done in section 3.2.2"

*"For all tests of background signals and blanks, the internal standard is added to the solutions at concentrations between 0.5-1 g/L in order generate aerosols of sufficient size for the AMS. This allows an analysis of any trace material present in the blank by creating an aerosol population to transfer the trace material into the AMS and allows for a background subtraction using the internal standard."*

**Line 170-172: Was the chamber run in a batch or continuous mode? Is the seed concentration of 60 µg/m3 the initial concentration in the chamber? What about the 500 ppbv ozone? Is that initial concentration as well? Do the concentrations vary over time? The filter samples were collected for 10 hr. How did the particle concentration vary over that 10 hr period? In Figure 5b, is the online AMS spectra averaged for that 10 hours?**

The chamber was run in constant-volume, semi-batch mode.  Further details on loadings have also been added:

"Chamber aerosol (enabling offline vs. online comparisons) was generated in the MIT 7.5-$m^3$ Teflon environmental chamber, *run in continuous-volume, "semi-batch" mode*.  Details on the facility are given elsewhere (Hunter et al., 2014). Experiments were run at 20 °C, < 5% RH, in the dark, and under low-$NO_x$ *(< 10 ppb)* conditions  using ozone as the oxidant.  Ammonium sulfate seeds were added for *an initial* concentration of ~60 $\mu g/m^3$.  The VOC, $\alpha$-pinene, had an initial mixing ratio of 100 ppb; a penray lamp (Jelight model 600) was used to *add an initial ozone concentration of  ~700 ppb* ozone. *The ozone concentration decreased due to consumption and dilution to 400 ppb by the end of the experiment.  The initial organic loading was ~70 $\mu g/m^3$ and decayed due to dilution, sampling, and wall loss to a final value of ~ 18 $\mu g/m^3$."*

Yes, in Figure 5b, the online AMS is the average spectra over that 10 hour period.  We have added text to section 3.3 clarifying this.

**Line 172-173, Line 178-179, and Line 324: It is not clear to me how the 10-min sampling can represent the "blank". What kind of blank? Before the start of the experiments, what do the filters collect? Can 10 min be enough? What do the authors mean "blank subtraction was carried out with a scaling of the**

**filter blank to 12% of the sample signal, as determined from the internal standard in each sample"? The internal standard is not mentioned in the previous text.**

The blank tested here is a filter blank that gives the background on any organic material present on the teflon filter as well as any organic material added during the sample preparation process. For blanks, the standard protocol is to carry out the same procedures as the sampling. Thus, placing the filter in the holder is more to test the contamination that comes from the lab and/or filter handling instruments (tweezers) than to actually sample anything in the filter holder itself. The filter holder is usually thoroughly cleaned between runs and we believe that 10 min is sufficient time to pick up any remaining, easily transferred, material in the filter itself. We have added text clarifying this to section 3.3:

"*These blanks provide the background for any trace organic material on the filters before collection as well as any background organic material added during sample preparation.*"

The blank subtraction was carried out by using the ratio of the organic to the internal standard for the blank and for the sample. The scaling factor was found to be 12% for this sample, thus the total intensity of the blank mass spectrum was multiplied by this value before subtracting it from the chamber filter. For these analysis, we spike every sample with ~0.5-1 g/L internal standard in order to enable this type of analysis. Text clarifying this has been added to the manuscript and is detailed in this document two comments above this one.

**Line 180-186: Although the filter samples are from another study, the authors should provide enough details about the sample/samples used in this study. What is the sampling period? "Blown down to dryness" means completely dry? I suspect completely dryness affects the semivolatile SOA species. If not, how concentrated the solution is used for SVN-AMS measurements compared to ambient loading.**

The time frame for the SOAS campaign is given in the experimental. The time frame for the sample shown in the manuscript is given in the section discussing that mass spectrum (section 3.3). This sample is a night sample collected on July 4 from 8 pm to 7 am the following day.

Yes, here blow down to dryness means completely dry. We also agree that this process will affect semivolatile SOA species. Thus, the volatility and the solubility of the compounds in the sample will likely affect the mass spectrum observed compared to ambient. We mention that the loss of volatile compounds during sample preparation will likely be a factor influencing the differences we observe in the text in section 3.3

**Line 190-191: The grade of the reagents should be provided.**

All reagents were 99% purity or more, text communicating this has been added.

**Line 198-199: How is this white power prepared for the SVN-AMS analysis? Dissolve into additional water or just melt?**

The DOM was dissolved in MilliQ water at ~1 g/L, this has been added to the experimental.

**Line 200: Please clarify for each case (chamber, SOAS, and DOM), how many fiters/samples were used in this study.**

We did not need to combine filters to generate sufficient mass for each sample and the DOM sample provided by our collaborators was sufficient to generate more than one mL of 1 g/L solution. Thus each mass spectrum shown in the figure is a data set from one filter sample or DOM sample. We have added text addressing this to the experimental:

*"For all analyses presented here (chamber and ambient) sufficient mass was extracted to enable the analysis of individual filter samples, with no combination of extracts from different samples required."*

**Line 240: The description about the variations should be more quantitative. Given such big variations, I think it is necessary to clearly state that this method is not good for quantitative analysis of the mass concentrations of the samples.**

The variations we observe are in the amount of material that is sampled into the AMS. We use the internal standard to account for these variations. The ratio of the analyte to the internal standard is consistent, despite variations in the total aerosol mass produced. The calibration curves in figure 4 demonstrate this fact. To quantify unknowns, known amounts of the internal standard are added to each sample. The ratio of the signals for the unknown to the internal standard can then be used, with the calibration curve, to calculate the unknown concentration by multiplying by the known concentration of the internal standard in the solution.

We have added text clarifying some additional details for how this type of analysis can be carried out for ambient samples in section 3.2.2:

*"For the calibration curve, the ratios of the AMS signals for the analyte over the internal standard are compared to the ratios for known solution concentrations, thus correcting any variations in the mass of analyte nebulized. For quantification of unknowns, known concentrations of the internal standard are added to the samples at ratios comparable to what is used for the calibration curve. The ratio of the measured AMS signals can then be used to calculate the unknown analyte concentration from the calibration curve."*

**Line 280-281: Where are the background coming from? If adding the internal standard to a sample solution, what happens?**

The background can come from many different sources. For the sample shown in Figure 4a it is likely trace material on the Kapton surface. There may also be trace material in the syringe used to load the sample. When the internal standard is added to the solution, we produce a solution that has enough material to form aerosols that can be measured in the AMS when the sample is nebulized. Text added section 3.3 clarifies this:

"*Typically*, the internal standards are added at the same order of magnitude concentration as the sample. *For all tests of background signals and blanks, the internal standard is added to the solutions at concentrations between 0.5-1 g/L in order to generate aerosols of sufficient size for the AMS. This allows an analysis of any trace material present in the blank by creating an aerosol population to transfer the trace material into the AMS and allows for a background subtraction using the internal standard*."

**Line 293-306: The internal standard method should be tested for ambient samples with the suggested RIE of 1.4; otherwise the goodness of this method remains unclear. Section 3.3: The run-to-run variations is large for nebulization efficiency. What about the run-to-run variations of elemental composition and mass spectra. Please provide.**

We thank the reviewer for this comment, testing the method with ambient samples is a current project that the first author is working on and is beyond the scope of this work. The use of an appropriate calibration standard is a challenge for any quantification of unknown organic mixtures. We recommend a slope of 1.4 as a starting point as that is the currently accepted RIE for atmospheric organic material.

We have clarified this by adding the following text to the end of section 3.2.2:

"For extracts of atmospheric aerosol or other smaller organic mixtures, the RIE of 1.4, which is typically used for AMS measurements (Canagaratna et al., 2007; Jimenez et al., 2016; Xu et al., 2018), is likely the best value to use *as an initial calibration slope*. For extracts of other types of organic mixtures, compounds that have a structure similar to the average organic composition should be used to calibrate the samples."

For the mass spectral analysis, the average mass spectrum across multiple injections is used for the analysis. This improves the S/N for the analysis and we have added text to the experimental clarifying this:

"For the work shown here, mass spectra were collected every 0.5 seconds for ~15-18 seconds in the "open" state, followed by 3 seconds in the closed state. The closed spectrum provides information on the instrument background, including contributions from gas phase species, and is subtracted from the open spectrum in data processing. *For the analysis of the mass spectrum and the elemental ratios, the average mass spectrum across all injections is used. For quantification, the total signal under each injection pulse (see below, Figure 2a) is used*."

**Line 316: The model of the TSI atomizer as well as the operating conditions should be provided.**

The atomizer is a TSI 3076 aerosol generator and the backing gas pressure was 40 psi. This information has been added to section 3.3.

**Line 320: I don't understand how the dot product (of what?) represents the similarity of the two spectra.**

The dot product used here is the dot product between the intensities of matching peaks in the two mass spectra. Larger dot products indicate a greater degree of similarity between the two mass spectra. We have added text clarifying this to the beginning of section 3.3:

"The degree of agreement can be described by the dot product *of the intensities for matching peaks in* the two spectra, as well as the log of the intensities before taking the dot product (log-dot product), which gives the lower intensity peaks greater weight."

**Line 344-353: The mass spectra showed in Figure 5c are indeed quite different in terms of many distinguished ions. I think saying the two mass spectra have a high degree of agreement (Line 344) or one is well represented by the other (Line 351-352) is improper and misleading. Since the major similarity appears for m/z 41-44, the authors may present some of the major peak ratios (e.g., f44-to-f43) instead. Also, in Table 1, the uncertainty of elemental ratios should be provided. Using f44 to derive O:C is associated with much greater uncertainty than the AMS method does. The comparison for Look Rock somewhat indicates that the elemental ratios are less sensitive to the method.**

We agree that the degree of overlap for the ambient sample is lower than what is observed for the chamber experiment.

The high degree of overlap we mention is characterized by the high dot product between the spectra (0.98). We qualify this by discussing the substantially larger variation in the lower intensity peaks and use the log-dot product to provide a better representation of this variation (0.90).

To clarify that the "high degree of overlap" is referring specifically to the dominant ion intensities and to discuss the limitations of the elemental ratios measured by this analysis we adding the following text:

"Additional work is necessary to quantify the importance of these effects *and care should be taken when comparing the full mass spectra for on-line compared to off-line SVN-AMS analysis*. The high degree of overlap *in the intensities of the dominant ions* between the online (AMS/ACSM) measurements and offline (SVN-AMS) results indicates that the ensemble organic composition for these aerosol samples is generally well-represented by the SVN-AMS measurements (Table 1). *However, the estimated elemental ratios from a lower resolution AMS are more uncertain than from the HR-ToF-AMS. Thus, the ratios for these samples in Table 1 are provided only as a demonstration of the overall agreement between the two techniques*."

**Line 355-356: Not only "remain in the condensed phase after nebulization" but also after intensive pretreatments (e.g., serial dilution/concentration etc. in Line 197-198).**

We have added the following text to address this comment:

"For the SVN, the small sample volume requirements can make it attractive for the analysis of other environmental samples that are soluble in water (or organic solvents) and that have low enough vapor pressures to remain in the condensed phase *during sample preparation and* after nebulization."

**Line 365-366: Why the pretreatments in Line 197-198 are needed for the SVN-AMS method for DOM? Tests on real water samples should be presented. The effects of dilution or concentration on the analysis deserve further discussions.**

The pretreatments for the DOM are necessary because the concentration of organic relative to salts in ocean water is ~ 30 x 10$^{-6}$. Thus, if pure ocean water is sampled, the dominant signals will be salts present in the sea water with very trace organic signal. The sample pre-treatment steps are standard steps carried out for the analysis of the high molecular weight fraction of DOM. We add the following text to clarify this at the end of section 3.3.

"*Here we demonstrate the analysis of the high molecular weight fraction of dissolved organic matter (DOM) with the SVN-AMS. The DOM sample was prepared using a standard protocol for the isolation of this fraction of the organic material (see section 2.3).* Figure 5d shows an example AMS mass spectrum from DOM *collected* from the Pacific Ocean.

**Technical Remarks:**

**Line 27: What kind of "diameter"? Also, to be clear, this diameter is for dry particles.**

The diameter measurements were made using an SMPS so they are electrical mobility diameters. This diameter is for particles that have dried during transit to the SMPS. Text clarifying both these topics has been added to the beginning of the results and discussion.

"The particles have size distributions centered at 150-200 nm *(electrical mobility diameter). These particles were sampled into the SMPS without passing through a drier. The SVN was approximately three meters further away from the inlet of the SMPS so the particles are likely to be somewhat smaller than those entering the AMS, due to water evaporation in the dry carrier gas.*"

**Line 29 and Line 281: Please add "material" after "organic".**

This has been added

**Figure 2: Please properly write the ions in the legend.**

This has been corrected

**Line 240: "Slowly" is an improper word here, maybe "slightly".**

This section has been reworded and that sentence has been removed.

**Figure 5: To be clear, the "SVN" in panels b), c), d) is indeed "SVN-AMS". In panel a), the legend of "TSI" should be "Atomizer". And the "Online" in panel c) should be "AMS"?**

Changes have been made.

**Line 307: "3.1" should be "3.3".**

This has been corrected.